# Clinical and Echocardiographic Predictors for the Presence of Late Gadolinium Enhancement on Cardiac Magnetic Resonance Imaging in Patients with Carbon Monoxide Poisoning

**DOI:** 10.3390/diagnostics14010060

**Published:** 2023-12-27

**Authors:** Dong-Hyuk Cho, Jung-Woo Son, Young In Kim, Jihye Lim, Ho-Sung Jeon, Sung Min Ko, Yong Sung Cha

**Affiliations:** 1Department of Cardiology, Korea University College of Medicine, Seoul 26426, Republic of Korea; why012@gmail.com; 2Division of Cardiology, Department of Internal Medicine, Yonsei University Wonju College of Medicine, Wonju 26493, Republic of Korea; soneycar@gmail.com (J.-W.S.); yinkim2002@gmail.com (Y.I.K.); nahuuu@yonsei.ac.kr (H.-S.J.); 3Department of Biostatistics and Center of Biomedical Data Science, Yonsei University Wonju College of Medicine, Wonju 26493, Republic of Korea; jihye8082@yonsei.ac.kr; 4Department of Radiology, Yonsei University Wonju College of Medicine, Wonju 26493, Republic of Korea; ksm9723@yonsei.ac.kr; 5Department of Emergency Medicine, Yonsei University Wonju College of Medicine, Wonju 26493, Republic of Korea; 6Research Institute of Hyperbaric Medicine and Science, Yonsei University Wonju College of Medicine, Wonju 26493, Republic of Korea

**Keywords:** prognosis, cardiac injury, echocardiography, magnetic resonance image, cohort study

## Abstract

Late gadolinium enhancement (LGE) on cardiac magnetic resonance imaging (CMRI) reflects the burden of myocardial damage in carbon monoxide (CO) poisoning. This study aimed to identify the clinical and echocardiographic parameters that can predict myocardial LGE on CMRI in CO poisoning. This prospective observational study included patients who presented with acute CO poisoning and elevated troponin I and underwent echocardiography and CMRI to identify myocardial damage at a tertiary university hospital between August 2017 and May 2019 and August 2020 and July 2022. Based on the CMRI findings, participants were categorized into LGE and non-LGE groups. The median age of the 155 patients was 51.0 years, and 98 (63.2%) were males. Median times from emergency department arrival to either CMRI or echocardiography were 3.0 days each. The LGE group included 99 (63.9%) patients with LGE positivity on CMRIs. Time from rescue to hyperbaric oxygen therapy >4 h (odds ratio (OR): 3.31, 95% confidence interval (CI): 1.28–8.56, *p =* 0.01); serum lactate levels >2 mmol/L (OR: 2.62, 95% CI: 1.20–5.73, *p =* 0.02); and left ventricular global longitudinal strain >−16% (OR: 2.95, 95% CI: 1.35–6.47, *p =* 0.007) were significant predictors of LGE positivity. The area under the curve of these predictors was 0.711. Our prediction model, which combines the clinical parameters with left ventricular global longitudinal strain, may be helpful in the early detection of LGE positivity.

## 1. Introduction

Approximately 50,000 patients in the United States are admitted to the emergency departments (EDs) annually for carbon monoxide (CO) poisoning, resulting in 1500 deaths [1,2]. Despite efforts, the colorless, tasteless, odorless, and non-irritating properties of CO make it difficult to prevent CO poisoning [3]. CO poisoning can lead to cardiac injuries [4]. Myocardial injury, defined as increased cardiac enzyme levels or ischemic changes indicated in electrocardiography, was observed in 37% of patients with CO poisoning who required hyperbaric oxygen therapy (HBO_2_) [4]. We had previously reported that 74.4% of patients with acute CO poisoning and elevated troponin I (TnI) developed CO-induced cardiomyopathy [5]. In a follow-up study [4], mortality was significantly higher among patients who experienced myocardial injury than among those who did not, and death from cardiovascular causes was more common among patients with myocardial injury [6]. Furthermore, in a nationwide cohort study, individuals with CO poisoning were at a higher risk of major adverse cardiovascular events than those without CO poisoning [7]; these findings indicate that myocardial injury is common in acute CO poisoning and increases long-term cardiovascular mortality. 

Cardiac magnetic resonance imaging (MRI) is a novel tool for detecting myocardial fibrosis via late gadolinium enhancement (LGE) detection [8,9,10,11]. Our recent prospective study reported that 69.2% of patients with CO poisoning, elevated TnI levels, and no underlying cardiovascular diseases had LGE on their cardiac MRI. The mid-wall injury was the most common [12]. The pathogenesis of the mid-wall injury may be via the inhibition of mitochondria [3,13,14], direct myocardial damage [3], global endothelial dysfunction, and increased production of free radicals [3,15,16]. Most chronic-phase images showed no interval change in patients who underwent follow-up cardiac MRI [12]. Mortality within one year was only observed in the group with LGE, suggesting that myocardial fibrosis assessed using LGE is associated with poor clinical prognosis [12]. However, cardiac MRI is not easily accessible to all patients with CO poisoning because of its cost, time intensiveness, need for intermittent breathing holds, and the use of contrast agents. Therefore, it would be desirable if the indications for cardiac MRI can be further specified when patients with a high likelihood of LGE positivity on cardiac MRI can be identified using simpler diagnostic tools. Patients at risk of LGE positivity could be screened using cardiac MRI and followed up by cardiologists, allowing for early intervention in the event of cardiac sequelae.

Global longitudinal strain (GLS) measured using speckle tracking echocardiography is an emerging novel tool that reflects subclinical myocardial damage [17]. GLS may detect myocardial damage earlier than left ventricular systolic dysfunction evaluated by measuring the left ventricular ejection fraction [18]. Therefore, we aimed to evaluate the clinical risk factors and conventional and two-dimensional (2D) speckle tracking echocardiographic parameters as predictors of LGE positivity on cardiac MRI in acute CO poisoning.

## 2. Materials and Methods

### 2.1. Study Design and Setting

The study data were derived from an independent cohort at single tertiary academic hospitals in South Korea. The study data were obtained from (1) our previous prospective observational study in which patients were enrolled between August 2017 and May 2019 (first period) (informed consent obtained; ClinicalTrials.gov identifier: NCT04419298) [12] and (2) cohort data (CARE CO cohort) collected prospectively between August 2020 and July 2022 (second period) (informed consent obtained; ClinicalTrials.gov identifier: NCT04490317). The institutional review board of Wonju Severance Christian Hospital (approval number: CR322073) approved this study, and the study protocol complied with the ethical guidelines of the Declaration of Helsinki.

The exclusion criteria for patients with CO poisoning were (1) age below 19 years; (2) TnI level not available or not elevated within 24 h; (3) cardiac arrest; (4) history of cardiovascular diseases, such as angina pectoris, myocardial infarction, coronary revascularization procedure, heart failure, or cardiomyopathy; (5) not undergoing transthoracic echocardiography (TTE) by a cardiologist or a cardiologist could not interpret their TTE; (6) not undergoing cardiac MRI due to creatinine clearance <30 mL/min, artificial retention devices, unable to cooperate, or a history of hypersensitivity to gadolinium; and (7) a radiologist could not interpret the cardiac MRI due to poor image quality. In this study, the data of 88 out of the 155 patients included in our previous study were analyzed [12]. All included patients received HBO_2_.

Acute CO poisoning was diagnosed based on the patient’s history and blood carboxyhemoglobin levels >5% (>10% for excessive smokers). Even if the blood carboxyhemoglobin level was normal, patients in whom a CO source was identified with related symptoms and signs were diagnosed as having CO poisoning. Myocardial damage was defined based on elevated TnI levels (reference range ≥450 pg/mL) measured within 24 h of ED arrival. A screening test conducted by the National Forensic Service (Wonju, Republic of Korea) was used to confirm the simultaneous intake of cardiotoxic drugs. Patients with CO poisoning received treatment with 100% oxygen therapy through a face mask with a reservoir bag. Patients with loss of consciousness, neuropsychological symptoms, such as severe headache, and signs such as dysarthria, extreme weakness, cognitive dysfunction, myocardial ischemia (ischemic electrocardiogram change or TnI elevation), cardiovascular dysfunction, severe acidosis, or >25% carboxyhemoglobin level were treated with HBO_2_. All included patients were treated with HBO_2_ as all manifestations were indications for HBO_2_ (myocardial ischemia or cardiovascular dysfunction, loss of consciousness, and neurological symptoms and signs). The same treatment protocol was used during the two study periods.

### 2.2. Study Variables

Clinical variables investigated included age, sex, intentionality, CO source, maximum estimated CO exposure time (hours), medical comorbidities (such as hypertension, diabetes mellitus, and hyperlipidemia), time from rescue to HBO_2_ (hours), Glasgow Coma Scale score at the scene or ED, loss of consciousness interval, shock, and time from ED admission to cardiac MRI and TTE (days). Laboratory variables investigated included carboxyhemoglobin, creatinine, creatine kinase, TnI, β-type natriuretic peptide, and arterial blood gas (bicarbonate and lactate), which were measured immediately after ED arrival. Serum TnI was measured within 24 h of ED arrival, and peak TnI was defined as the highest value among those measured.

CO poisoning-related mortality and neurocognitive outcomes were investigated using the global deterioration scale (GDS; stages: 1–7) [19], ranging 1–7 points. Patients at GDS stages 1–7 experienced impairment that ranged from none (stage 1) to loss of motor skills and all language skills except for inaudible unintelligible sounds (stage 7). Mortality was evaluated until 12 months after CO poisoning.

### 2.3. Cardiac MRI

We obtained cardiac MRI delay enhancement images using a Magnetom Skyra 3T system (Siemens Healthineers, Erlangen, Germany) with a 32-channel body coil. We administered 0.2 mmol/kg gadoteridol (ProHance; Bracco S.p.A., Milan, Italy) via a vein to obtain LGE images. We used a phase-sensitive inversion recovery sequence 10 min after injection (echo time: 1.99 ms, repeat time: 695 ms, flip angle: 20°, slice thickness: 8 mm, field of view: 37 × 37 cm, acquisition matrix size: 256 × 192, and pixel size: 1.5 × 1.9 mm). The inversion time was adjusted for each patient using the look-locker sequence to achieve a null signal from the normal myocardium. Cardiac MRI was performed within 7 days of exposure to CO for all included patients and 4 to 5 months after, if follow-up was possible. A certified radiologist (S.M.K), blinded to clinical information, interpreted the cardiac MRI twice (2 months apart between the first and second). Based on cardiac MRI findings, patients were classified into LGE and non-LGE groups.

### 2.4. Echocardiographic Variables

Patients with elevated serum TnI levels were subjected to 2D echocardiography to assess cardiac function within 7 days of ED visits. Experienced cardiologists performed echocardiography using a Vivid E9 (General Electrical Medical System, Horton, Norway) with a 2.5-MHz probe. We obtained conventional echocardiographic images using parasternal long- and short-axis views and apex four-chamber and two-chamber views. LV and left atrial chamber sizes were quantified, and LV ejection fraction was measured using the biplane Simpson method [20]. Mitral inflow velocity was obtained in the apical four-chamber view using pulsed-wave Doppler echocardiography during early (E) and late filling (A). The early diastolic (e′) and peak systolic (s′) mitral and annular velocities of the septal mitral and lateral mitral annuli were evaluated using tissue Doppler imaging, and the results were averaged [21]. Cardiologists specializing in cardiac imaging (D.H.C and J.W.S), blinded to clinical and cardiac MRI data, interpreted the echocardiographic findings. In speckle tracking strain analysis, a second harmonic B-mode image of the standard apical four-chamber, three-chamber, and two-chamber LV views was acquired at the end-systolic frame and stored digitally in a cine loop form. The LV GLS was analyzed using an ECHOPAC software program (Version 204.41.4; General Electrical Medical System, Horton, Norway) and calculated as the average value of the longitudinal peak negative strain in 17 segments.

### 2.5. Study Outcome

This study aimed to evaluate the predictors of LGE positivity on cardiac MRI during the acute phase of CO poisoning (within 7 days).

### 2.6. Statistical Analysis

Continuous variables were reported as means ± standard deviations or medians (interquartile range), whereas categorical variables were reported as frequencies (percentages). The normality of continuous variables was assessed using the Shapiro–Wilk test. The differences in characteristics between the LGE groups were compared using the independent two-sample t-tests or Mann–Whitney U test for continuous variables and chi-square or Fisher’s exact test for categorical variables.

Based on the receiver operating characteristic (ROC) curve, continuous variables were dichotomized in the logistic regression analysis. The optimal cut-off point was calculated using the maximum Youden index. Multivariate logistic regression was performed to determine the adjusted effects of predictors, and the model included age, sex, and statistically significant variables in the univariate logistic regression. Based on a significance level of 0.05, backward elimination was used to select optimal predictors. The prediction model’s performance was evaluated using the area under the ROC curve (AUC). 

All statistical analyses were performed using SAS statistical software version 9.4 (SAS Institute, Cary, NC, USA) and R version 4.1.2 (R Core Team, Vienna, Austria). Statistical significance was set at *p* < 0.05.

## 3. Results

### 3.1. Patient Characteristics

During this study, a total of 812 patients visited the hospital with acute CO poisoning. Patients aged under 19 years were excluded (first period: 24 patients; second period: 26 patients). During the first period, 164 patients with normal TnI were excluded, and during the second period, 194 patients with TnI not checked or normal TnI were excluded. A total of 156 patients underwent cardiac MRI and 1 patient whose cardiac MRI could not be interpreted was excluded. Finally, 155 patients (first period: 88 patients; second period: 67 patients) were included (Figure 1). Informed consent was obtained from all included patients.

Table 1 shows the baseline characteristics of the study participants. The median age was 51.0 years, and 98 patients (63.2%) were males. Charcoal (76.8%) was the most common source of CO poisoning. The median maximum estimated CO exposure time was 8.0 h. All patients received HBO_2_; the median time from rescue to HBO_2_ was 8.0 h. The median times from ED arrival to either cardiac MRI or echocardiography were 3.0 days, each. The median initial and peak TnI levels were 545 and 1146 pg/mL, respectively. Among included patients, the 99 (63.9%) patients with LGE positivity on cardiac MRI were classified as the LGE group. Bicarbonate (*p* = 0.02) and lactate (*p* = 0.02) levels differed significantly between the two groups in the context of LGE. In the univariate logistic regression, after the continuous variables were dichotomized, significant differences were observed in the time from rescue to HBO_2_ (*p* = 0.027), creatine kinase level (*p* = 0.045), initial and peak TnI levels (*p* = 0.037 and 0.049), β-type natriuretic peptide levels (*p* = 0.024), bicarbonate levels (*p* = 0.032), and lactate levels (*p* = 0.011) (Appendix A). No significant between-group differences in mortality and GDS were observed. However, five mortality cases occurred in the LGE group during the 1-year follow-up after CO poisoning (Table 1).

Table 2 shows the echocardiographic results. CO-induced cardiomyopathy was observed in eight (5.2%) patients, and a Takotsubo-like pattern and global dysfunction in six (3.9%) and eight (5.2%) patients, respectively. The median LV ejection fraction and LV GLS were 60.0% and −16.9%, respectively. In echocardiographic parameters, the median LV GLS showed a significant decrease in movement in the LGE group when compared with that in the non-LGE group (−16.3% vs. −17.4%, *p* = 0.01) (Figure 2). Right ventricular (RV) systolic function represented by the RV fractional area change was not different between groups. In the univariate logistic regression, after the continuous variables were dichotomized, significant differences were observed in LV ejection fraction (*p* = 0.03), LV GLS (*p* = 0.002), LV end-systolic volume (*p* = 0.01), LV end-systolic dimension (*p* = 0.02), and LV mass index (*p* = 0.04) (Appendix A).

### 3.2. Main Results

In the multivariate logistic regression analysis performed with the clinical and echocardiographic parameters, time from rescue to HBO_2_ > 4 h (odds ratio (OR) 3.31, 95% confidence interval (CI) 1.28–8.56, *p* = 0.01); serum lactate level > 2 mmol/L (OR 2.62, 95% CI 1.20–5.73, *p* = 0.02); and LV GLS > −16% (OR 2.95, 95% CI 1.35–6.47, *p* = 0.007) were significant predictors for LGE positivity (Table 3). The AUC of the combined three predictors (reduced model) was 0.711 (Figure 3). The AUC of only LV GLS and the full model including all variables were 0.630 and 0.779, respectively (Figure 3). 

## 4. Discussion

While cardiac MRI is considered the gold standard for evaluating myocardial characteristics, it can be costly and challenging to perform in uncooperative patients during the acute phase. Therefore, the primary objective of this study was to investigate the role of clinical and echocardiographic parameters in predicting the presence of LGE in cases of acute CO poisoning. In this study, we investigated the utility of clinical risk factors and 2D speckle tracking echocardiographic parameters in predicting LGE positivity using cardiac MRI in acute CO poisoning. The results revealed that LV ejection fraction was not different between the LGE and non-LGE groups, and LV GLS was more impaired in the LGE group. Traditional risk factors, including time from rescue to HBO_2_ and serum lactate levels, were associated with LGE positivity. LV GLS was the only significant predictor of LGE positivity among the echocardiographic parameters in the multivariate logistic regression analysis. These findings suggest that 2D speckle tracking echocardiography is a feasible and useful tool for identifying patients with CO poisoning with a high risk of LGE positivity in cardiac MRI.

The correlation between LGE positivity on cardiac MRI and the presence of myocardial fibrosis has been well established in various studies on cardiovascular diseases [10,11]. It is challenging to precisely determine whether myocardium with LGE indicates fibrosis or myocardial edema in CO poisoning; however, in our previous study [12], we observed that 67.6% of the 37 patients who underwent follow-up cardiac MRI showed no interval change at 4–5 months. This lack of change in myocardial appearance, as indicated by LGE, may reflect chronic myocardial damage, such as myocardial fibrosis, rather than acute reactive change such as myocardial edema. We believe that to enhance tissue characterization, additional CMR variables such as T1, T2, and extracellular volume (ECV) are necessary. In this study, we only enrolled patients with acute CO poisoning having elevated TnI levels and included a middle-aged population without significant cardiovascular diseases; however, 99 patients (63.9%) exhibited LGE positivity on cardiac MRI, indicating a high prevalence of myocardial injury in acute CO poisoning. 

LV ejection fraction is a surrogate marker of LV systolic function in various clinical situations [22]. However, LV ejection fraction has multifaceted limitations. First, there is observer dependency in LV ejection fraction measurement, and second, accuracy highly depends on image quality and the observer’s skill and experience [23]. Additionally, a non-linear relationship exists between an extremely high LV ejection fraction and clinical outcomes. An ejection fraction of 50% or higher is considered normal; however, evidence suggests that patients with LV ejection fractions >75% may have worse clinical outcomes, such as an increased risk of heart failure and sudden cardiac death [24]. Another limitation of LV ejection fraction is that it may not detect the early changes in cardiac function because it measures global systolic function and does not consider regional changes in myocardial function. In contrast, strain imaging, which measures myocardial deformation, can detect subtle changes in regional function before changes in ejection fraction become apparent [25,26].

Our findings suggest that LV GLS may be a more sensitive and specific measure of subclinical myocardial damage in acute CO poisoning patients than LV ejection fraction. The impaired LV GLS in the LGE group indicates its ability to detect myocardial damage that LV ejection fraction could not detect. However, further studies are required to confirm these findings and determine the clinical significance of LV GLS in this population. The usefulness of LV GLS for the early detection of myocardial damage has been studied in various cardiovascular diseases. For instance, LV GLS demonstrated a higher prognostic value for mortality than LV ejection fraction in acute decompensated heart failure [26]. Furthermore, in chemotherapy-induced cardiotoxicity, early reduction in LV GLS during chemotherapy preceded a drop in LV ejection fraction or overt heart failure [27], suggesting that LV GLS is a more useful parameter for the early prediction of cardiotoxicity. Our findings add to the growing evidence that LV GLS is a useful parameter for the early detection of cardiovascular diseases, including CO-induced cardiotoxicity. 

In this study, the time from rescue to HBO_2_ and serum lactate levels at the ED were the predictors of LGE positivity among parameters that are not related to echocardiography. Serum lactate, a marker of tissue ischemia, correlates with the patients’ neurologic outcomes and is a useful prognostic factor in acute CO poisoning [28]. This may also apply to cardiac damage, suggesting that tissue ischemia is an important parameter in determining the burden of myocardial fibrosis in acute CO poisoning. An increase in the interval from rescue to HBO_2_ increases the risk of a poor outcome [29]. In our previous study, patients who received HBO_2_ within 6 h of CO exposure had a better 6-month neurocognitive prognosis than those treated within 6–24 h [29]. However, no proven evidence suggests that HBO_2_ reduces myocardial damage and fibrosis in patients with acute CO poisoning. Further studies are required to determine whether HBO_2_ or antifibrotic therapy can reduce the burden of myocardial damage in these patients. In this study, carboxyhemoglobin level was not a predictor of LGE positivity. We thought that carboxyhemoglobin level may not provide accurate information on CO exposure because the time to measurement and prior oxygen therapy can result in dramatic changes from the levels at the scene. Therefore, we should not make clinical decisions based on its value.

Patients with CO poisoning may have neurological problems, making it difficult to perform cardiac MRI on them, despite MRI being the most sensitive tool for detecting myocardial damage. Therefore, there is an urgent need to develop prediction models that use convenient tools to detect myocardial damage at an early stage. Our prediction model, which employs clinical parameters and LV GLS, may be useful for stratifying patients with a high risk of developing future myocardial damage. This could lead to early consultation with a cardiologist and prompt intervention to reduce myocardial damage in patients with acute CO poisoning. 

Ours was a prospective study investigating the association between LV GLS and myocardial injury in patients with acute CO poisoning. However, this study had some limitations. First, it was conducted in a single center, which may limit the generalizability of our findings to other populations. Second, this study only included patients with elevated TnI levels and no history of significant cardiovascular diseases, which may have limited the range of our findings. Third, there was no information on the LGE pattern before CO poisoning. Fourth, LV GLS was a more sensitive marker of myocardial damage than LV ejection fraction; however, how these findings can be translated into clinical practice remains unclear. Further studies are required to determine the clinical significance of these findings and whether an early LV GLS-based intervention can improve outcomes in patients with acute CO poisoning. Fifth, the source of CO for the majority of the included patients was charcoal. Sixth, it is important to note that the direct association between myocardial fibrosis and the presence of LGE on cardiac MRI has not been conclusively established. As we did not have access to the pre-exposure echocardiographic or cardiac MRI data, the certainty of our findings in relation to CO exposure is somewhat reduced. Therefore, future studies must be conducted to validate and confirm the potential association between myocardial fibrosis and LGE in the context of CO exposure. Lastly, the sample size was relatively small, and studies using larger sample sizes are required to confirm our findings.

In conclusion, our prediction model, which combines clinical parameters with LV GLS, can facilitate early detection of LGE positivity in patients with acute CO poisoning, leading to early consultation with a cardiologist.

## Figures and Tables

**Figure 1 diagnostics-14-00060-f001:**
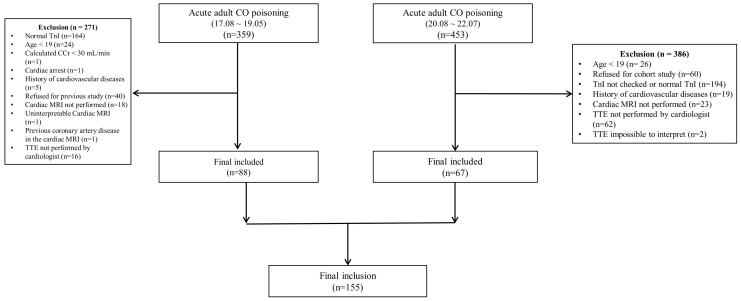
Study flow chart. CO = carbon monoxide poisoning, TnI = troponin I, CCr = creatine clearance, MRI = magnetic resonance imaging, TTE = transthoracic echocardiography.

**Figure 2 diagnostics-14-00060-f002:**
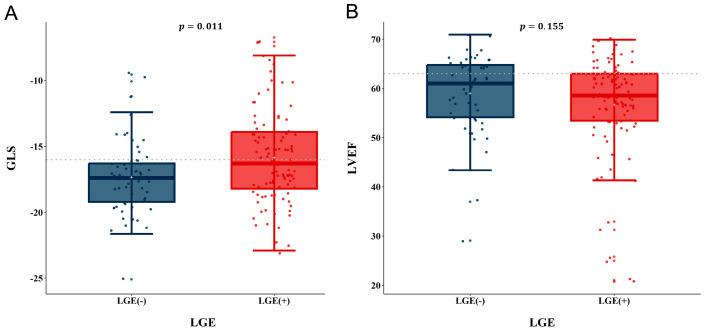
Comparison of LV GLS (**A**) and LV ejection fraction (**B**) between the non-LGE and LGE groups. LV GLS = left ventricle global longitudinal strain, LGE = late gadolinium enhancement, LV = left ventricle.

**Figure 3 diagnostics-14-00060-f003:**
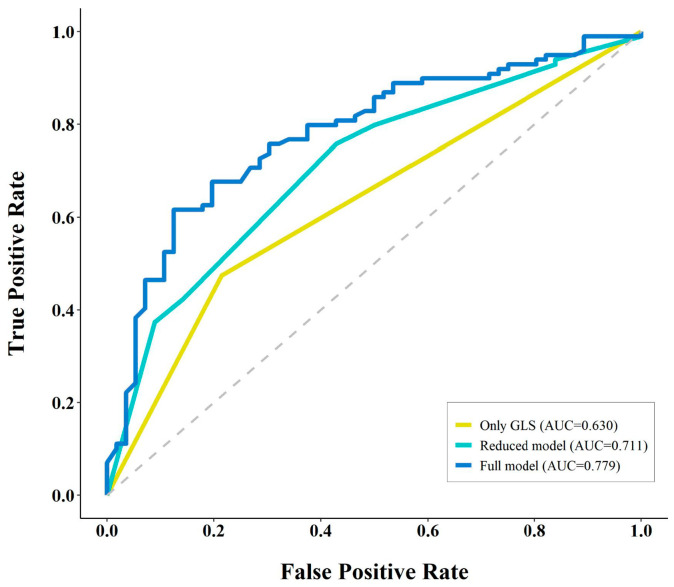
AUC of the predictive models. AUC = area under the receiver operating characteristic curve, GLS = global longitudinal strain.

**Table 1 diagnostics-14-00060-t001:** Baseline characteristics of the total cohort.

Variables	Total (*n* = 155)	Non-LGE (*n* = 56, 36.1%)	LGE (*n* = 99, 63.9%)	*p*-Value
Age (years)	51.0 (38.0–64.0)	52.5 (37.0–64.5)	49.0 (38.0–62.0)	0.710
Sex (male)	98 (63.2)	33 (58.9)	65 (65.7)	0.404
Intentionality	86 (55.5)	31 (55.4)	55 (55.6)	0.981
Source				0.639
Charcoal	119 (76.8)	42 (75.0)	77 (77.8)	
Gas and oil	26 (16.8)	9 (16.1)	17 (17.2)	
Fire	10 (6.5)	5 (8.9)	5 (5.1)	
Maximum estimated CO exposure times (hours)	8.0 (1.7–8.0)	4.5 (1.4–8.0)	8.0 (2.0–9.0)	0.203
Comorbidities				
Hypertension	38 (24.5)	14 (25.0)	24 (24.2)	0.916
Diabetes mellitus	15 (9.7)	6 (10.7)	9 (9.1)	0.743
Hyperlipidemia	9 (5.8)	5 (8.9)	4 (4.0)	0.286
Time from rescue to HBO_2_ (hours)	8.0 (4.8–12.3)	8.5 (4.2–12.0)	7.4 (5.0–12.5)	0.705
Glasgow Coma Scale	8.0 (8.0–12.0)	9.5 (7.0–12.0)	8.0 (8–12)	0.892
Loss of consciousness	142 (91.6)	51 (91.1)	91 (91.9)	1.000
Shock	20 (12.9)	7 (12.5)	13 (13.1)	0.910
Laboratory findings				
COHb (%)	26.7 (12.0–39.3)	29.5 (11.6–39.7)	26.7 (12.0–38.5)	0.888
Creatinine (mg/dL)	0.9 (0.8–1.2)	0.9 (0.8–1.1)	1.0 (0.8–1.2)	0.154
Creatine kinase (U/L)	372 (148–2336)	318 (180–1238)	529 (143–3041)	0.324
Troponin I (pg/mL)				
Initial	545 (128–2688)	375 (103–2462)	1010 (156–2933)	0.097
Peak	1146 (189–3439)	653 (185–2758)	1616 (214–3940)	0.157
BNP (pg/mL)	55 (14–136)	43 (15–94)	68 (13–198)	0.285
Bicarbonate (mmol/L)	20.2 (17.3–22.8)	21.3 (19–23.5)	19.8 (16.5–22.4)	0.017
Lactate (mmol/L)	3.1 (1.9–5.1)	2.4 (1.7–4.7)	3.4 (2.1–5.2)	0.023
Time from ED arrival to CMRI (days)	3.0 (1.0–4.0)	3.0 (2.0–4.0)	3.0 (1.0–4.0)	0.435
Time from ED arrival to TTE (days)	3.0 (2.0–4.0)	2.0 (2.0–4.0)	3.0 (2.0–4.0)	0.063
Outcomes				
GDS at 1 month	2.0 (1.0–4.0)	2.0 (1.0–4.0)	2.0 (1.0–4.0)	0.994
GDS at 6 months	1.0 (1.0–4.0)	2.0 (1.0–4.0)	1.0 (1.0–3.0)	0.481
GDS at 1 year	1.0 (1.0–3.0)	1.5 (1.0–4.0)	1.0 (1.0–3.0)	0.385
Mortality within 1 year	5 (4.2)	0 (0.0)	5 (6.3)	0.168

Data are expressed as frequencies (percentages), medians (interquartile ranges), or means ± standard deviations. The mortality of 120 patients is evaluated. GDS is evaluated to assess the neuropsychological performance and activities of daily living in 133, 103, and 89 patients with GDS at 1, 6, and 12 months, respectively. LGE = late gadolinium enhancement, CO = carbon monoxide, HBO_2_ = hyperbaric oxygen therapy, COHb = carboxyhemoglobin, BNP = β-type natriuretic peptide, ED = emergency department, CMRI = cardiac magnetic resonance imaging, TTE = transthoracic echocardiography, GDS = global deterioration scale.

**Table 2 diagnostics-14-00060-t002:** Baseline transthoracic echocardiographic characteristics of the total cohort.

Variables	Total (*n* = 155)	Non-LGE (*n* = 56, 36.1%)	LGE (*n* = 99, 63.9%)	*p*-Value
Presence of CO-induced CMP	8 (5.2)	2 (3.6)	6 (6.1)	0.712
Takotsubo-like pattern	6 (3.9)	1 (1.8)	5 (5.1)	0.419
Global dysfunction	8 (5.2)	2 (3.6)	6 (6.1)	0.712
TTE parameters				
LV EF (%)	60.0 (53.9–63.9)	61.0 (54.1–64.8)	58.6 (53.3–63.0)	0.155
LV GLS (%)	−16.9 (−18.5–−14.6)	−17.4 (−19.3–−16.3)	−16.3 (−18.2–−13.8)	0.011
LVEDV (mL)	103.4 ± 29.6	102.9 ± 29.6	103.7 ± 29.7	0.863
LVESV (mL)	42.0 (33.2–51.0)	37.4 (32.3–51.5)	44.0 (35.0–51.0)	0.175
LVEDD (mm)	50.8 ± 4.4	50.4 ± 4.6	51.1 ± 4.3	0.368
LVESD (mm)	34.0 (31.0–37.0)	33.0 (30.5–37.0)	34.0 (31.0–37.0)	0.428
SWT (mm)	8.0 (7.0–9.0)	8.0 (7.0–8.0)	8.0 (7.0–9.0)	0.934
PWT (mm)	8.0 (7.0–9.0)	8.0 (7.0–8.0)	8.0 (7.0–9.0)	0.474
LVMI (g/m^2^)	81.1 ± 18.3	78.6 ± 19.3	82.4 ± 17.7	0.210
LAVI (mL/m^2^)	28.0 (22.0–33.0)	28.0 (21.0–33.0)	28.0 (22.0–33.0)	0.940
Septal e′ (cm/s)	8.0 (6.0–11.0)	8.0 (5.5–11.0)	8.0 (6.0–11.0)	0.736
E/e′ (ratio)	9.0 (7.0–11.0)	9.0 (8.0–11.5)	9.0 (7.0–11.0)	0.342
RVSP (mmHg)	27.4 ± 7.9	27.2 ± 7.6	27.6 ± 8.2	0.819
RV end-diastolic area	20.0 (16.0–24.0)	20.0 (16.0–23.5)	20.0 (16.0–24.0)	0.788
RV end-systolic area	12.0 (9.0–14.0)	11.5 (10.0–14.0)	12.0 (9.0–14.0)	0.852
RV FAC	40.4 ± 4.6	40.1 ± 4.3	40.5 ± 4.8	0.623
Basal RVD	38.0 (34.0–43.0)	38.5 (34.0–43.5)	38.0 (34.0–43.0)	0.838
Mid RVD	32.0 (29.0–37.0)	32.5 (28.5–37.0)	32.0 (29.0–37.0)	0.736
RV long-axis dimension	64.0 (58.0–74.0)	65.0 (60.0–74.0)	64.0 (57.0–73.0)	0.620

Data are expressed as frequencies (percentages), medians (interquartile ranges), or means ± standard deviations. LGE = late gadolinium enhancement, CO = carbon monoxide, CMP = cardiomyopathy, TTE = transthoracic echocardiography, LV EF = left ventricular ejection fraction, LV GLS = left ventricular global longitudinal strain, LVEDV = left ventricular end-diastolic volume, LVESV = left ventricular end-systolic volume, LVEDD = left ventricular end-diastolic dimension, LVESD = left ventricular end-systolic dimension, SWT = septal wall thickness, PWT = posterior wall thickness, LVMI = left ventricular mass index, LAVI = left atrial volume index, Septal e′ = septal early mitral tissue velocity, RVSP = right ventricular systolic pressure, RV: right ventricular, RV FAC: RV fractional area change, RVD: RV dimension.

**Table 3 diagnostics-14-00060-t003:** Predictors of LGE positivity using multivariate logistic regression.

Variables	Univariate		Multivariate, Reduced Model		
	Standardized β	OR (95% CI)	Standardized β	Adjusted OR (95% CI)	*p*-Value
Sex					
Male		Ref.			
Female	−0.077	0.75 (0.38–1.47)			
Age (years)					
≤45		Ref.			
>45		0.70 (0.35–1.38)			
Time from rescue to HBO_2_ (hours)					
≤4		Ref.		Ref.	
>4		2.418 (1.001–5.841)	0.239	3.31 (1.28–8.56)	0.014
Laboratory findings					
Creatine kinase (U/L)					
≤500		Ref.			
>500	0.190	1.99 (1.02–3.91)			
Initial troponin I (pg/mL)					
>480		Ref.			
≤480		0.49 (0.25–0.96)			
Peak troponin I (pg/mL)					
>990					
≤990	−0.184	0.514 (0.264–0.999)			
BNP (pg/mL)					
>95		Ref.			
≤95		0.43 (0.20–0.90)			
Bicarbonate (mmol/L)					
>20		Ref.			
≤20		2.08 (1.07–4.07)			
Lactate (mmol/L)					
≤2		Ref.		Ref.	
>2		2.56 (1.24–5.29)	0.237	2.62 (1.20–5.73)	
LV EF (%)					
>63		Ref.			
≤63		2.23 (1.08–4.6)			
LV GLS (%)					
≤−16		Ref.		Ref.	
>−16		3.31 (1.57–7.02)		2.95 (1.35–6.47)	0.007
LVESV (mL)					
≤43		Ref.			
>43	0.234	2.34 (1.18–4.61)			
LVESD (mm)					
≤29		Ref.			
>29		3.21 (1.17–8.84)			
LVMI (g/m^2^)					
≤70		Ref.			
>70		2.14 (1.05–4.35)			

Only variables derived from predictive factors with a significance ≤ 0.05 in Appendix A are included. LGE = late gadolinium enhancement, OR = odds ratio, CI = confidence interval, BNP = β-type natriuretic peptide, LV EF = left ventricular ejection fraction, LV GLS = left ventricular global longitudinal strain, LVESV = left ventricular end-systolic volume, LVESD = left ventricular end-systolic dimension, LVMI = left ventricular mass index.

## Data Availability

Data will be made available by the authors upon reasonable request and with permission.

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
