# Peer review of "Clinical and Echocardiographic Predictors for the Presence of Late Gadolinium Enhancement on Cardiac Magnetic Resonance Imaging in Patients with Carbon Monoxide Poisoning"

_diagnostics, 2023, doi:10.3390/diagnostics14010060_

Round 1

Reviewer 1 Report

Comments and Suggestions for Authors

Congratulations to the authors and their team for conducting this very interesting research. I appreciated the quality of the evidence, the well structured article and the results. I consider this work can be maximized by a future article following up these patients and seeing the impact of the LGE presence on MACE. I also appreciate the recognition of all the limitations of the study. 

I only have some minor comments: 

- lines 98-101 need rephrasing

- line 164 - GLS is calculated in 17 segments

- it is okay to conclude on LGE presence at 3 days post exposure, but not on fibrosis/necrosis (for example, during an acute myocardial infarction, it takes weeks/months until all the myocardial edema is gone so we can properly assess the consequences in terms of fibrosis) - so I suspect that the LGE present at 3 days is mostly myocardial edema. I consider this point should be discussed in the discussion part

- also, if available, data on parametric mapping values (T1, T2 or even STIR) would be interesting, since a high T2 would confirm the presence of edema and would indicate the need for rescanning these patients after several months (as we do for myocarditis).

Comments on the Quality of English Language

- lines 98-101 need rephrasing

Author Response

Responses to the reviewer’s comments

We would like to thank the editor and reviewers for their valuable feedback and suggestions to improve our manuscript titled "Clinical and echocardiographic predictors for the presence of late gadolinium enhancement on cardiac magnetic resonance imaging in patients with carbon monoxide poisoning".

Based on the reviewers' comments, we have revised the manuscript accordingly. We have addressed each point raised by the reviewers and provided our responses below. The suggested changes have been implemented and are highlighted in red in the revised manuscript that is provided along with these responses.

Reviewer 1

Comments and Suggestions for Authors

Congratulations to the authors and their team for conducting this very interesting research. I appreciated the quality of the evidence, the well structured article and the results. I consider this work can be maximized by a future article following up these patients and seeing the impact of the LGE presence on MACE. I also appreciate the recognition of all the limitations of the study.

I only have some minor comments:

- lines 98-101 need rephrasing

[Response] We have rephrased these sentences for clarity and correctness (Materials and Methods: “Study design and setting”, pages 2–3, lines 96–99).

- line 164 - GLS is calculated in 17 segments

[Response] We have now changed the sentence according to your comment (Materials and Methods: “Echocardiographic variables”, page 4, line 162).

- it is okay to conclude on LGE presence at 3 days post exposure, but not on fibrosis/necrosis (for example, during an acute myocardial infarction, it takes weeks/months until all the myocardial edema is gone so we can properly assess the consequences in terms of fibrosis) - so I suspect that the LGE present at 3 days is mostly myocardial edema. I consider this point should be discussed in the discussion part.

- also, if available, data on parametric mapping values (T1, T2 or even STIR) would be interesting, since a high T2 would confirm the presence of edema and would indicate the need for rescanning these patients after several months (as we do for myocarditis).

[Response] We appreciate the reviewer for this important inquiry. It is challenging to precisely determine whether myocardium with LGE indicates fibrosis or myocardial edema as suggested. However, our previous research results indicated minimal changes during the follow-up CMRI, which may suggest myocardial fibrosis. To enhance tissue characterization, additional CMRI variables, such as T1, T2, and extracellular volume (ECV), are necessary. When enrolling patients, obtaining such supplementary information was difficult due to the original CMRI protocol's limitations, particularly in patients with altered consciousness or poor cooperation. With the advent of new CMRI equipment, analyzing these additional variables within a shorter timeframe is now feasible. New patients are undergoing evaluations that include these variables, facilitating a more comprehensive assessment. Please find our modifications within the text:

“The correlation between LGE positivity on cardiac MRI and the presence of myocardial fibrosis has been well-established in various studies on cardiovascular diseases [10,11]. It is challenging to precisely determine whether myocardium with LGE indicates fibrosis or myocardial edema in CO poisoning; however, in our previous study [12], we observed that 67.6% of the 37 patients who underwent follow-up cardiac MRI showed no interval change at 4-5 months. This lack of change in myocardial appearance, as indicated by LGE, may reflect chronic myocardial damage, such as myocardial fibrosis, rather than acute reactive change such as myocardial edema. We believe that to enhance tissue characterization, additional CMR variables such as T1, T2, and extracellular volume (ECV) are necessary.”

(Discussion, pages 10–11, lines 283–292)

Reviewer 2 Report

Comments and Suggestions for Authors

Cho et al approached a rather interesting topic: the assessment of various clinical and echocardiographic predictors for the presence of late gadolinium enhancement in patients with CO poisoning. The manuscript is decently written, the statistics are accurate, however there are some aspects that should be further addressed:

Major comments:

Please further detail the mechanism of the CO-induced fibrosis pattern described in introduction section (mid-wall) and correlate it with the GLS findings of the included patients (decreased strain of the septum?). This is important, as it may reveal a specific multimodal imagery pattern, such as in the case of amyloidosis or hypertrophic CM.

The study totally lacks data concerning the right ventricle. The RV (longitudinal) dysfunction is an important negative predictor, especially in acute settings involving cardiorespiratory axis (i.e. CO intoxication). Please provide at least basic RV parameters, such as TAPSE or S’. Even better if supported by MRI data.

The lack of medical history concerning conditions that may evolve with myocardial fibrosis is a serious limitation in the reproducibility of the results.

Please provide additional references concerning the mechanism of CO-induced myocardial dysfunction.

Minor comments:

Why did the authors not mention the actual hospital where the study was conducted?

Review by a native English speaker as some phrases are oddly conceived.

Best regards,

The Reviewer

Comments on the Quality of English Language

Review by a native English speaker as some phrases are oddly conceived.

Author Response

Responses to the reviewer’s comments

We would like to thank the editor and reviewers for their valuable feedback and suggestions to improve our manuscript titled "Clinical and echocardiographic predictors for the presence of late gadolinium enhancement on cardiac magnetic resonance imaging in patients with carbon monoxide poisoning".

Based on the reviewers' comments, we have revised the manuscript accordingly. We have addressed each point raised by the reviewers and provided our responses below. The suggested changes have been implemented and are highlighted in red in the revised manuscript that is provided along with these responses.

Reviewer 2

Comments and Suggestions for Authors

Cho et al approached a rather interesting topic: the assessment of various clinical and echocardiographic predictors for the presence of late gadolinium enhancement in patients with CO poisoning. The manuscript is decently written, the statistics are accurate, however there are some aspects that should be further addressed:

Major comments:

  1. Please further detail the mechanism of the CO-induced fibrosis pattern described in introduction section (mid-wall) and correlate it with the GLS findings of the included patients (decreased strain of the septum?). This is important, as it may reveal a specific multimodal imagery pattern, such as in the case of amyloidosis or hypertrophic CM.

[Response] We appreciate the reviewer for this important query. We think the pathogenesis of mid-wall injury may be inhibition of mitochondria, direct myocardial damage, global endothelial dysfunction, and increased production of free radicals. We have added this to the Introduction. In response to the reviewer's suggestion, additional analysis was conducted to examine whether regional strain was reduced among the patients in whom LGE was observed in the septum. The difference between GLS and regional strain values was defined as delta strain to analyze whether regional strain was relatively decreased. However, regional strain values were not different based on the location of LGE (R-table). Further research is necessary to explore the concordance between LGE patterns and regional strain.

 R-Table. Comparison of segmental delta strain

ΔBS strain

ΔMS strain

ΔAS strain

ΔBL strain

ΔML strain

ΔAL strain

Total (n=29)

-2.8±0.5

0.89±0.6

2.5±1.0

-0.8±0.5

-0.6±0.6

0.3±1.0

LGE (septal) (n=21, 72%)

-2.5±0.7

1.0±8

2.4±1.3

-1.4±0.7

-1.0±0.7

-0.7±1.3

LGE (lateral)

(n=8, 28%)

-3.6±0.7

0.7±0.6

2.7±1.4

0.8±0.4

0.6±1.0

2.8±1.0

“The pathogenesis of the mid-wall injury may be via the inhibition of mitochondria [1-3], direct myocardial damage [2], global endothelial dysfunction, and increased production of free radicals [2,4,5].” (Introduction, page 2, lines 59–61)

  1. The study totally lacks data concerning the right ventricle. The RV (longitudinal) dysfunction is an important negative predictor, especially in acute settings involving cardiorespiratory axis (i.e. CO intoxication). Please provide at least basic RV parameters, such as TAPSE or S’. Even better if supported by MRI data.

[Response] We agree with the reviewer on this important point. In the evaluation of acute cardiotoxicity, RV dysfunction and its recovery are one of the significant predictors for the short- and long-term prognosis. Unfortunately, our echocardiography laboratory only evaluates RV longitudinal function represented by TAPSE or RV S’. Instead of these parameters, we have added the parameters of RV chamber size and RV fractional area change. Between the groups with and without LGE, these RV parameters were neither different nor increased the predictive value for the presence of LGE. We have added this in the revised manuscript (Results, pages 7–8, line 238).

Table 2. Baseline transthoracic echocardiographic characteristics of the total cohort.

Variables

Total

Non-LGE

LGE

P-value

(n=155)

(n=56, 36.1%)

(n=99, 63.9%)

Presence of CO-induced CMP

8 (5.2)

2 (3.6)

6 (6.1)

0.712

 Takotsubo-like pattern

6 (3.9)

1 (1.8)

5 (5.1)

0.419

 Global dysfunction

8 (5.2)

2 (3.6)

6 (6.1)

0.712

TTE parameters

 LV EF (%)

60.0 (53.9 – 63.9)

61.0 (54.1 – 64.8)

58.6 (53.3 – 63.0)

0.155

 LV GLS (%)

-16.9 (-18.5 – -14.6)

-17.4 (-19.3 – -16.3)

-16.3 (-18.2 – -13.8)

0.011

 LVEDV (mL)

103.4 ± 29.6

102.9 ± 29.6

103.7 ± 29.7

0.863

 LVESV (mL)

42.0 (33.2 – 51.0)

37.4 (32.3 – 51.5)

44.0 (35.0 – 51.0)

0.175

 LVEDD (mm)

50.8 ± 4.4

50.4 ± 4.6

51.1 ± 4.3

0.368

 LVESD (mm)

34.0 (31.0 – 37.0)

33.0 (30.5 – 37.0)

34.0 (31.0 – 37.0)

0.428

 SWT (mm)

8.0 (7.0 – 9.0)

8.0 (7.0 – 8.0)

8.0 (7.0 – 9.0)

0.934

 PWT (mm)

8.0 (7.0 – 9.0)

8.0 (7.0 – 8.0)

8.0 (7.0 – 9.0)

0.474

 LVMI (g/m2)

81.1 ± 18.3

78.6 ± 19.3

82.4 ± 17.7

0.210

 LAVI (mL/m2)

28.0 (22.0 – 33.0)

28.0 (21.0 – 33.0)

28.0 (22.0 – 33.0)

0.940

 Septal e′ (cm/s)

8.0 (6.0 – 11.0)

8.0 (5.5 – 11.0)

8.0 (6.0 – 11.0)

0.736

 E/e′ (ratio)

9.0 (7.0 – 11.0)

9.0 (8.0 – 11.5)

9.0 (7.0 – 11.0)

0.342

 RVSP (mmHg)

27.4 ± 7.9

27.2 ± 7.6

27.6 ± 8.2

0.819

 RV end diastolic area (cm2)

20.0 (16.0 – 24.0)

20.0 (16.0 – 23.5)

20.0 (16.0 – 24.0)

0.788

 RV end systolic area (cm2)

12.0 (9.0 – 14.0)

11.5 (10.0 – 14.0)

12.0 (9.0 – 14.0)

0.852

 RV FAC (%)

40.4 ± 4.6

40.1 ± 4.3

40.5 ± 4.8

0.623

 Basal RVD (mm)

38.0 (34.0 – 43.0)

38.5 (34.0 – 43.5)

38.0 (34.0 – 43.0)

0.838

 Mid RVD (mm)

32.0 (29.0 – 37.0)

32.5 (28.5 – 37.0)

32.0 (29.0 – 37.0)

0.736

 RV long-axis dimension (mm)

64.0 (58.0 – 74.0)

65.0 (60.0 – 74.0)

64.0 (57.0 – 73.0)

0.620

RV: right ventricular, RVFAC: RV fractional area change, RVD: RV dimension

“Right ventricular (RV) systolic function represented by the RV fractional area change was not different between groups.” (Results, page 7, lines 228–230)

  1. The lack of medical history concerning conditions that may evolve with myocardial fibrosis is a serious limitation in the reproducibility of the results.

[Response] We agree with the reviewer on this insightful comment. The lack of information on the LGE pattern before CO poisoning is a significant limitation. Addressing this issue may necessitate a community-based CMR study. We have added this as a limitation in the revised manuscript:

“Third, there was no information on the LGE pattern before CO poisoning.” (Discussion, page 12, lines 352–353).

  1. Please provide additional references concerning the mechanism of CO-induced myocardial dysfunction.

[Response] We have added additional references (13, 14, 15, and 16) in the Introduction (page 2, lines 59–61).

Minor comments:

  1. Why did the authors not mention the actual hospital where the study was conducted?

[Response] We have now added the name of the actual hospital (Wonju Severance Christian Hospital) (Materials and Methods: “Study Design and Setting”, page 2, line 87).

  1. Review by a native English speaker as some phrases are oddly conceived.

[Response] We have ensured a thorough proofreading of the manuscript by a native English proofreading organization.

Round 2

Reviewer 2 Report

Comments and Suggestions for Authors

The authors have addressed my previous suggestions in a professional manner, with a significant improvement of the manuscript.